# Modelling, Analysis, and Simulation of the Micro-Doppler Effect in Wideband Indoor Channels with Confirmation Through Pendulum Experiments

**DOI:** 10.3390/s20041049

**Published:** 2020-02-14

**Authors:** Ahmed Abdelgawwad, Alireza Borhani, Matthias Pätzold

**Affiliations:** Faculty of Engineering and Science, University of Agder, P.O. Box 509, 4898 Grimstad, Norway; alireza.borhani@uia.no (A.B.); matthias.paetzold@uia.no (M.P.)

**Keywords:** spectrogram, 3D no n-stationary channels, indoor channels, doppler frequency, channel state information, Wi-Fi 802.11n, inertial measurement units, micro-doppler effect, CSI

## Abstract

This paper is about designing a 3D no n-stationary wideband indoor channel model for radio-frequency sensing. The proposed channel model allows for simulating the time-variant (TV) characteristics of the received signal of indoor channel in the presence of a moving object. The moving object is modelled by a point scatterer which travels along a trajectory. The trajectory is described by the object’s TV speed, TV horizontal angle of motion, and TV vertical angle of motion. An expression of the TV Doppler frequency caused by the moving scatterer is derived. Furthermore, an expression of the TV complex channel transfer function (CTF) of the received signal is provided, which accounts for the influence of a moving object and fixed objects, such as walls, ceiling, and furniture. An approximate analytical solution of the spectrogram of the CTF is derived. The proposed channel model is confirmed by measurements obtained from a pendulum experiment. In the pendulum experiment, the trajectory of the pendulum has been measured by using an inertial-measurement unit (IMU) and simultaneously collecting CSI data. For validation, we have compared the spectrogram of the proposed channel model fed with IMU data with the spectrogram characteristics of the measured CSI data. The proposed channel model paves the way towards designing simulation-based activity recognition systems.

## 1. Introduction

In wireless communications, the compound Doppler effect caused by the moving objects or bodies opened up opportunities for many applications. These applications track the scattered wave components by the moving bodies for drone detection [1], gesture recognition [2], human gait assessment for diagnosis and rehabilitation [3], and tracking human activities using no n-wearable radio-frequency-based (RF-based) elder-care [4]. Such waves contain the micro-Doppler effects corresponding to the moving bodies.

In channel modelling, the Doppler effect caused by moving scatterers has been modelled in two-dimensional (2D) stationary fixed-to-mobile radio in [5]. Then, this model has been extended for 2D no n-stationary fixed-to-fixed (F2F) indoor channels by considering the time-variant (TV) speed of the moving scatterer, angle of motion, angle of arrival, and angle of departure [6]. Later on, the TV Doppler frequency caused by the moving scatterer has been incorporated in three-dimensional (3D) channels by taking into account the TV azimuth angles of motion (AAOM), elevation angle of motion (EAOM), azimuth angle of departure (AAOD), elevation angle of departure (EAOD), azimuth angle of arrival (AAOA), and elevation angle of arrival (EAOA) for fixed-to-fixed channel models [6,7] and vehicle-to-vehicle channels [8]. To reveal the TV Doppler power characteristics of non-stationary multicomponent signals, a time-frequency distribution such as the spectrogram can be employed. The authors of [9] distinguished between aided and unaided gaits by means of the spectrogram. In [10] the angular velocities and lengths of rotating blades have been estimated by using the spectrogram. The spectrogram has been employed in gesture recognition [11] and human activity recognition (HAR) [12]. It has been used for the detection of gait asymmetry in [3,13], distinguishing between armed and unarmed persons for security services [14], and fall detection [15,16,17], as well.

The authors of [18] developed a software tool that captures the complex channel state information (CSI) of 30 subcarriers corresponding to orthogonal-frequency-division-multiplexing (OFDM). This software tool is compatible with commercial devices equipped with Intel NIC 5300 network interface cards and operates on the Wi-Fi 802.11n protocol [19]. An overview of studies on signal recognition, action recognition, and activity recognition by utilizing the amplitude of the measured CSI can be found in [20]. One of the challenges faced while processing CSI data to extract the micro-Doppler signatures is that the phases of such data are distorted as the transmitter and receiver are not clock synchronized [21,22,23,24,25,26,27,28]. To overcome this issue, one of the attempts is to apply the principle component analysis (PCA) [29,30,31,32] on the magnitude of the complex CSI data to denoise it, then to apply a one-sided spectrogram on the denoised data to reveal the positive frequency components of the spectrogram. Another attempt has been applied by using a phase sanitization technique [32,33,34] by employing linear transformation operation on the distorted phases of the 30 subcarriers. Such an operation gives a better pattern of the transformed phases. These attempts do not help to reveal the true Doppler power characteristics of the preprocessed CSI data. In [35] the phase distortions of CSI data have been eliminated by using a back-to-back (B2B) connection between the transmitter and receiver stations. This approach allows for revealing the true Doppler power characteristics of the measured CSI data.

The micro-Doppler effect of pendulum motion in bistatic and monostatic radar systems has been investigated in [36,37]. In [36], the micro-Doppler effect was analyzed by means of the one-sided spectrogram of both simulated radar signal and verified with the experiment. According to the best of the authors’ knowledge, there are no studies on the micro-Doppler effect of a swinging pendulum on measured calibrated CSI with B2B connection. There are no simultaneous conducted measurements by using the CSI tool and an IMU attached to the swinging pendulum, as well.

The Fresnel zone diffraction model has been described in [38,39,40,41]. Such a model has been used for CSI-based human activity recognition [42], human respiration sensing [43], and indoor human detection [44]. The Fresnel zone model is an envelope model that does not contain any phase information. The phase information is important for the analysis of the micro-Doppler signatures, which is our main focus in this paper. The motivation of this paper is to present a non-stationary wideband F2F channel model that has Doppler power characteristics similar to experimental data. According to the best of the authors’ knowledge, RF-based HAR, gesture recognition, and fall detection systems are designed based on experimental data, i.e., the machine used for detection or classification is trained by using experimental data. Thus, proposing such a model will help for simulation-based activity recognition systems by using it for training instead of using experimental data. This approach is time efficient and cost-effective. Instead of wasting too much time for collecting RF data for training, one can generate data sets of different scenarios by using lab simulations. In order to design such a realistic channel model with Doppler power characteristics close to experimental data, the TV trajectory of a moving object plays an important role as the Doppler shift depends on the TV speed, AAOM, EAOM, EAOD, AAOD, EAOA, and AAOA. The trajectory of the moving object can be measured by using an IMU which captures linear accelerations, and Euler angles simultaneously. The Euler angles are used to rotate the measured linear accelerations from the frame of the IMU to the reference frame. Then the rotated linear accelerations can be integrated to obtain TV velocities and displacements (trajectories). The TV velocities and displacements suffer from linear and quadratic drifts, respectively. In [45,46], these drift problems were addressed by employing zero update algorithms.

In this paper, we design a 3D no n-stationary wideband channel model for activity recognition. We model a‘moving object as a moving scatterer. The expressions for the TV speed, AAOM, EAOM, EAOD, AAOD, EAOA, and AAOA corresponding to the moving scatterer are presented. Then, the expression for the TV Doppler frequency caused by the moving scatterer is provided. Furthermore, the instantaneous channel phase of the moving scatterer and the complex channel transfer function (CTF) of the no n-stationary F2F channels are determined. Next, an approximate solution of the spectrogram of the complex CTF is presented to provide insight into the TV Doppler power characteristics of this model. We perform measurements of the calibrated CSI and IMU, simultaneously during a moving object experiment. The CSI data is calibrated by using a B2B connection to eliminate the TV phase distortions. Then, we feed the proposed channel model with the measured trajectory using the IMU. Finally, we show that the spectrogram of the calibrated measured CSI data and the channel model are matching. The TV mean Doppler shifts computed from both spectrograms are matching as well. The proposed channel model paves the way towards the design of software-RF-based human activity recognition and fall detection systems.

The rest of the paper is structured as follows. Section 2 exhibits the 3D multipath propagation scenario with moving and fixed objects. Section 3 presents the expressions of the TV channel parameters and the complex channel transfer function. The approximate analytical solution of the spectrogram of the complex channel function is provided in Section 4. Section 5 discusses the numerical and measurement results. The conclusion and future work are discussed in Section 6.

## 2. The 3D Geometrical Model

We consider the 3D geometrical model of a 3D multipath propagation channel shown in Figure 1. This figure shows a fixed Wi-Fi transmitter $T_{x}$ and a fixed Wi-Fi receiver $R_{x}$ which operate according to the IEEE 802.11n standard [19] with carrier frequency $f_{0}=5.32$ GHz and bandwidth B=20 MHz. The positions of $T_{x}$ and $R_{x}$ are denoted by $\left(x^{T},y^{T},z^{T}\right)$ and $\left(x^{R},y^{R},z^{R}\right)$, respectively. A moving object whose center of mass (CoM) is modelled for simplicity by a single moving (point) scatterer $S^{M}$ initially located at $\left(x_{M},y_{M},z_{M}\right)$. The trajectory of the moving scatterer $S^{M}$ is described by a TV velocity vector ${\vec{v}}_{M}\left(t\right)$ which can be expressed by the TV speed $v_{M}\left(t\right)$, the TV AAOM $\alpha_{v_{M}}\left(t\right)$, and the TV EAOM $\beta_{v_{M}}\left(t\right)$. Each fixed object is modelled by a fixed scatterer $S_{m}^{F}$ (▴) for m=1,2,⋯,M, where M denotes the number of fixed scatterers (objects). The TV parameters $\beta_{M}^{T}\left(t\right)$, $\alpha_{M}^{T}\left(t\right)$, $\beta_{M}^{R}\left(t\right)$, and $\alpha_{M}^{R}\left(t\right)$ designate the TV EAOD, TV AAOD, TV EAOA, and TV AAOA, respectively. We assume single-bounce scattering, i.e., each wave that is launched from $T_{x}$ is bounced by either a fixed scatterer $S_{m}^{F}$ or a moving scatterer $S^{M}$ before arriving at $R_{x}$.

## 3. The Channel Transfer Function

The TV velocity vector ${\vec{v}}_{M}\left(t\right)$ of the moving scatterer $S^{M}$ is presented as
(1)$${\vec{v}}_{M}\left(t\right)={\left[\begin{array}{c} v_{M,x}\left(t\right)\;\;,\;\;v_{M,y}\left(t\right)\;\;,\;\;v_{M,z}\left(t\right) \end{array}\right]}^{T}$$
where ${\left[\cdot \right]}^{T}$ denotes the vector transpose operation. The velocities $v_{M,x}\left(t\right)$, $v_{M,y}\left(t\right)$, and $v_{M,z}\left(t\right)$ can be expressed in terms of the TV speed $v_{M}\left(t\right)$, TV EAOM $\beta_{v_{M}}\left(t\right)$, and TV AAOM $\alpha_{v_{M}}\left(t\right)$ as
(2)$$\begin{array}{ccc} v_{M,x}\left(t\right) & = & v_{M}\left(t\right)\cos\left(\beta_{v_{M}}\left(t\right)\right)\cos\left(\alpha_{v_{M}}\left(t\right)\right) \end{array}$$
(3)$$\begin{array}{ccc} v_{M,y}\left(t\right) & = & v_{M}\left(t\right)\cos\left(\beta_{v_{M}}\left(t\right)\right)\sin\left(\alpha_{v_{M}}\left(t\right)\right) \end{array}$$
(4)$$\begin{array}{ccc} v_{M,z}\left(t\right) & = & v_{M}\left(t\right)\sin\left(\beta_{v_{M}}\left(t\right)\right) \end{array}$$
where
(5)$$\begin{array}{ccc} \alpha_{v_{M}}\left(t\right) & = & \mathrm{atan}2\left(v_{M,y}\left(t\right),v_{M,x}\left(t\right)\right) \end{array}$$
(6)$$\begin{array}{ccc} \beta_{v_{M}}\left(t\right) & = & \arcsin\left(\frac{v_{M,z}\left(t\right)}{\sqrt{v_{M,x}^{2}\left(t\right)+v_{M,y}^{2}\left(t\right)+v_{M,z}^{2}\left(t\right)}}\right). \end{array}$$

The function atan2(·) in (Equation 5) represents the four-quadrant inverse trigonometric tangent function that provides an azimuth angle $\alpha_{v_{M}}\left(t\right)$ ranging from −π to π, unlike the regular arctan(·) function that provides an angle ranging from −π/2 to π/2. Note that the elevation angle $\beta_{v_{M}}\left(t\right)$ is within the range from −π/2 to π/2. By using the components of the TV velocity vector ${\vec{v}}_{M}\left(t\right)$ in (Equation 2)–(4), one can compute the displacements $x_{M}\left(t\right)$, $y_{M}\left(t\right)$, and $z_{M}\left(t\right)$ of the moving scatterer $S^{M}$ as
(7)$$\begin{array}{ccc} x_{M}\left(t\right) & = & x_{M}+\int_{0}^{t}v_{M,x}\left(t^{\prime }\right)dt^{\prime } \end{array}$$
(8)$$\begin{array}{ccc} y_{M}\left(t\right) & = & y_{M}+\int_{0}^{t}v_{M,y}\left(t^{\prime }\right)dt^{\prime } \end{array}$$
(9)$$\begin{array}{ccc} z_{M}\left(t\right) & = & z_{M}+\int_{0}^{t}v_{M,z}\left(t^{\prime }\right)dt^{\prime }. \end{array}$$

From the displacements in (7)–(9), the TV Euclidean distance $d_{M}^{T}\left(t\right)$ between the transmitter $T_{x}$ and the moving scatterer $S^{M}$ can be computed by
(10)$$d_{M}^{T}\left(t\right)=\sqrt{{\left(x_{M}\left(t\right)-x^{T}\right)}^{2}+{\left(y_{M}\left(t\right)-y^{T}\right)}^{2}+{\left(z_{M}\left(t\right)-z^{T}\right)}^{2}}.$$

Analogously, the Euclidean distance $d_{M}^{R}\left(t\right)$ between the receiver $R_{x}$ and the moving scatterer $S^{M}$ is given by
(11)$$d_{M}^{R}\left(t\right)=\sqrt{{\left(x_{M}\left(t\right)-x^{R}\right)}^{2}+{\left(y_{M}\left(t\right)-y^{R}\right)}^{2}+{\left(z_{M}\left(t\right)-z^{R}\right)}^{2}}.$$

By using the expressions of the displacements in (7)–(9) and the distances in (Equation 10) and (Equation 11), the TV EAOD $\beta_{M}^{T}\left(t\right)$, TV AAOD $\alpha_{M}^{T}\left(t\right)$, TV EAOA $\beta_{M}^{R}\left(t\right)$, and TV AAOA $\alpha_{M}^{R}\left(t\right)$ can be computed as follows: (12)$$\begin{array}{ccc} \beta_{M}^{T}\left(t\right) & = & \arcsin\left(\frac{z_{M}\left(t\right)-z^{T}}{d_{M}^{T}\left(t\right)}\right) \end{array}$$(13)$$\begin{array}{ccc} \alpha_{M}^{T}\left(t\right) & = & \mathrm{atan}2\left(y_{M}\left(t\right)-y^{T},\;x_{M}\left(t\right)-x^{T}\right) \end{array}$$(14)$$\begin{array}{ccc} \beta_{M}^{R}\left(t\right) & = & \arcsin\left(\frac{z_{M}\left(t\right)-z^{R}}{d_{M}^{R}\left(t\right)}\right) \end{array}$$(15)$$\begin{array}{ccc} \alpha_{M}^{R}\left(t\right) & = & \mathrm{atan}2\left(y_{M}\left(t\right)-y^{R},\;x_{M}\left(t\right)-x^{R}\right) \end{array}$$
where $\alpha_{M}^{T}\left(t\right),\;\alpha_{M}^{R}\left(t\right)\in \left(-\pi ,\pi \right]$ and $\beta_{M}^{T}\left(t\right),\;\beta_{M}^{R}\left(t\right)\in \left[-\pi /2,\pi /2\right]$. The TV propagation delay $\tau_{M}\left(t\right)$ of the propagation path from $T_{x}$ via $S^{M}$ to $R_{x}$ is given by
(16)$$\begin{array}{c} \tau_{M}\left(t\right)=\frac{d_{M}^{T}\left(t\right)+d_{M}^{R}\left(t\right)}{c_{0}}. \end{array}$$

In (Equation 16), the parameter $c_{0}$ denotes the speed of light. The CTF $H(t,\Delta f^{\left(q\right)})$ is given by
(17)$$\begin{array}{ccc} H\left(t,\Delta f^{\left(q\right)}\right) & = & H_{M}\left(t,\Delta f^{\left(q\right)}\right)+\sum_{m=1}^{M}H_{F,m} \end{array}$$
where
(18)$$\begin{array}{ccc} H_{M}\left(t,\Delta f^{\left(q\right)}\right) & = & c_{M}\;e^{j\left[\theta_{M}-2\pi \left(f_{0}+\Delta f^{\left(q\right)}\right)\tau_{M}\left(t\right)\right]} \end{array}$$
(19)$$\begin{array}{ccc} H_{F,m} & = & c_{F,m}\;e^{j\theta_{F,m}}. \end{array}$$

The superscript *q* in (Equation 17) represents the subcarrier index of OFDM communication systems that follows the IEEE 802.11n standard [19]. The parameter $\Delta f^{\left(q\right)}$ in (Equation 17) designates the subcarrier frequency which is given by
(20)$$\begin{array}{c} \Delta f^{\left(q\right)}=q\cdot \Delta \end{array}$$
for q∈{−28,−26,⋯,−2,−1,1,3,⋯,27,28}. In (Equation 20), the parameter Δ represents the subcarrier frequencies difference which has a value of 312.5 kHz [19]. The function $H_{M}\left(t,\Delta f^{\left(q\right)}\right)$ designates the complex CTF of the moving scatterer $S^{M}$ and the parameter $H_{F,m}$ denotes the complex CTF corresponding the *m*th fixed scatterer $S_{m}^{F}$. The expression in (Equation 17) is similar to the one in [47] [Equation (Equation 21)]. The only difference is that the multipath effect associated with the fixed scatterers is taken into account by adding the second term in (Equation 17). The first term in (Equation 17) designates the TV part of the CTF corresponding to the moving scatterer $S^{M}$ with a fixed path gain $c_{M}$ and stochastic phase process $\theta_{M}-2\pi \left(f_{0}+\Delta f^{\left(q\right)}\right)\tau_{M}\left(t\right)$ associated with the *q*th subcarrier [see (Equation 18)]. The second term in (Equation 17) is time-invariant and represents the sum of the M received multipath components corresponding to the M fixed scatterers. Each component of the second term in (Equation 17) is characterized by a fixed path gain $c_{m,F}$ and a random phase variable $\theta_{m,F}$ due to the interaction with the *m*th fixed scatterer $S_{m}^{F}$ [see (19)]. It should be mentioned that the phases $\theta_{M}$ and $\theta_{m,F}$ are identically and independently distributed (i.i.d), each follows a uniform distribution over −π and π, i.e., $\theta_{M},\;\theta_{m,F}∼U\left(-\pi ,\pi \;\right]$. The model presented in (Equation 17) is a stochastic model of the TV CTF $H(t,\Delta f^{\left(q\right)})$. The TV Doppler shift $f_{M}^{\left(q\right)}\left(t\right)$ of the moving scatterer $S^{M}$ and associated with the *q*th subcarrier index is expressed by using (Equation 16) in combination with the relationship $f_{M}^{\left(q\right)}\left(t\right)=-\left(f_{0}+\Delta f^{\left(q\right)}\right){\dot{\tau }}_{M}\left(t\right)$, which can be found in [47] [Equation (Equation 22)] as [48]
(21)$$\begin{array}{cc} f_{M}^{\left(q\right)}\left(t\right) & =-f_{\max}^{\left(q\right)}\left(t\right)\{\cos\left(\beta_{v_{M}}\left(t\right)\right)\;[\cos\left(\beta_{M}^{T}\left(t\right)\right)\;\cos\left(\alpha^{T}\left(t\right)-\alpha_{v_{M}}\left(t\right)\right)\\ \; & +\;\cos\left(\beta_{M}^{R}\left(t\right)\right)\;\cos\left(\alpha_{v_{M}}\left(t\right)-\alpha_{M}^{R}\left(t\right)\right)]+\sin\left(\beta_{v_{M}}\left(t\right)\right)\left[\;\sin\left(\beta_{M}^{T}\left(t\right)\right)+\sin\left(\beta_{M}^{R}\left(t\right)\right)\right]\} \end{array}$$
where function $f_{\max}^{\left(q\right)}\left(t\right)$ designates the maximum Doppler shift caused by the moving scatterer $S^{M}$ which is given by
(22)$$\begin{array}{c} f_{\max}^{\left(q\right)}\left(t\right)=\frac{\left(f_{0}+\Delta f^{\left(q\right)}\right)v_{M}\left(t\right)}{c_{0}}. \end{array}$$

From the expression in (Equation 21) and the relationship $f_{M}^{\left(q\right)}\left(t\right)=-\left(f_{0}+\Delta f^{\left(q\right)}\right){\dot{\tau }}_{M}\left(t\right)$, one can conclude that if the moving scatterer $S^{M}$ moves away from the $T_{x}$ and $R_{x}$ vicinity, the TV propagation delay $\tau_{M}\left(t\right)$ and its slope ${\dot{\tau }}_{M}\left(t\right)$ increase and the Doppler effect $f_{M}\left(t\right)$ has negative values, and vice versa. To obtain an approximate solution for the spectrogram of the CTF $H(t,\Delta f^{\left(q\right)})$ that will be discussed in Section 4, the Doppler frequency $f_{M}^{\left(q\right)}\left(t\right)$ in (Equation 21) can be approximated by *L* linear piecewise functions according to
(23)$$\begin{array}{c} f_{M}^{\left(q\right)}\left(t\right)\approx \;f_{M,l}^{\left(q\right)}\left(t\right)=f_{M}^{\left(q\right)}\left(t_{l}\right)+k_{M,l}^{\left(q\right)}\left(t-t_{l}\right) \end{array}$$
for $t_{l}<t\leq \;t_{l+1}$ and l=0,1,⋯,L−1, where $k_{M,l}^{\left(q\right)}$ denotes the slope of the approximated Doppler frequency $f_{M}^{\left(q\right)}\left(t\right)$ which is given by
(24)$$\begin{array}{ccc} k_{M,l}^{\left(q\right)} & = & \frac{f_{M}^{\left(q\right)}\left(t_{l+1}\right)-f_{M}^{\left(q\right)}\left(t_{l}\right)}{t_{l+1}-t_{l}}. \end{array}$$

The difference between two consecutive time instances $t_{l+1}$ and $t_{l}$, i.e., $\delta =t_{l+1}-t_{l}$ is the same for all values of l=0,1,⋯,L−1.

The TV mean Doppler shift $B_{f^{\left(q\right)}}^{\left(1\right)}\left(t\right)$ of the proposed 3D channel model can be computed by using (Equation 21) as [49]
(25)$$\begin{array}{c} B_{f^{\left(q\right)}}^{\left(1\right)}\left(t\right)=\frac{c_{M}^{2}\;f_{M}^{\left(q\right)}\left(t\right)}{c_{M}^{2}+\sum_{m=1}^{M}c_{F,m}^{2}}. \end{array}$$

The expression in (Equation 25) denotes the squared path gain $c_{M}^{2}$ multiplied by the Doppler frequency caused by the moving scatterer $f_{M}^{\left(q\right)}\left(t\right)$ divided by the sum of the squared path gain of all of the scatterers. Note that, if the sum of the squared path gains $\sum_{m=1}^{M}c_{F,m}^{2}$ is much less than the squared path gain of the moving scatterer $c_{M}^{2}$ of the moving scatterer $S^{M}$, i.e., $\sum_{m=1}^{M}c_{F,m}^{2}\ll c_{M}^{2}$, then the TV mean Doppler shift $B_{f^{\left(q\right)}}^{\left(1\right)}\left(t\right)$ will have values closer to those of the Doppler frequency of the moving scatterer $f_{M}^{\left(q\right)}\left(t\right)$, i.e., $B_{f^{\left(q\right)}}^{\left(1\right)}\left(t\right)\to f_{M}^{\left(q\right)}\left(t\right)$.

## 4. Spectrogram Analysis

In this paper, we employ the spectrogram approach [50] to reveal the TV Doppler power spectrum of the proposed channel model. The spectrogram $S_{H^{\left(q\right)}}\left(f,t\right)$ of the CTF $H(t,\Delta f^{\left(q\right)})$ corresponding to the *q*th subcarrier index is computed in three steps. First, a sliding window w(t) is multiplied by the CTF $H(t,\Delta f^{\left(q\right)})$. In this paper, we choose a Gaussian window function [50] [Equation (2.3.1)]
(26)$$w\left(t\right)=\frac{1}{\sqrt{\sigma_{w}\sqrt{\pi }}}e^{-\frac{t^{2}}{2\sigma_{w}^{2}}}$$
where the parameter $\sigma_{w}$ is called the Gaussian window spread. The window function w(t) is real, positive, and even. It has a no rmalized energy, i.e., $\int_{-\infty }^{\infty }w^{2}\left(t\right)=1$. By multiplying the window function w(t) by the CTF $H(t,\Delta f^{\left(q\right)})$, the short-time CTF $x_{H^{\left(q\right)}}\left(t^{\prime },t\right)$ is obtained as
(27)$$x_{H^{\left(q\right)}}\left(t^{\prime },t\right)=H\left(t^{\prime },\Delta f^{\left(q\right)}\right)w\left(t^{\prime }-t\right)$$
where the variables *t* and $t^{\prime }$ are the local time and the running time, respectively. The second step is to compute the short-time Fourier transform (STFT) $X_{H^{\left(q\right)}}\left(f,t\right)$ of $x_{H^{\left(q\right)}}\left(t^{\prime },t\right)$. By using the approximation of the TV Doppler shift provided in (Equation 23), the STFT $X_{H^{\left(q\right)}}\left(f,t\right)$ associated with the *q*th subcarrier is obtained as
(28)$$\begin{array}{lll} X_{H^{\left(q\right)}}\left(f,t\right) & = & \int_{-\infty }^{\infty }x_{H^{\left(q\right)}}\left(t^{\prime },t\right)e^{-j2\pi ft^{\prime }}dt^{\prime }\\ & \approx & \frac{e^{-j2\pi ft}}{\sqrt{\sigma_{w}}\pi^{1/4}}\left\{H_{M}\left(t,\Delta f^{\left(q\right)}\right)\;G\left(f,f_{M,l}^{\left(q\right)}\left(t\right),\sigma_{x,M,l}^{2}\right)+\sum_{m=1}^{M}H_{F,m}\;G\left(f,0,\sigma_{x,F,m}^{2}\right)\right\} \end{array}$$
for $t_{l}<t\leq \;t_{l+1}$$\left(l=0,1,\cdots ,L-1\right)$, where
(29)$$\begin{array}{ccc} G\left(f,f_{M,l}^{\left(q\right)}\left(t\right),\sigma_{x,M,l}^{2}\right) & = & \frac{1}{\sqrt{2\pi }\sigma_{x,M,l}}e^{-\frac{{\left(f-f_{M,l}^{\left(q\right)}\left(t\right)\right)}^{2}}{2\sigma_{x,M,l}^{2}}} \end{array}$$
(30)$$\begin{array}{ccc} \sigma_{x,M,l}^{2} & = & \frac{1-j2\pi \sigma_{w}^{2}k_{M,l}^{\left(q\right)}}{{\left(2\pi \sigma_{w}\right)}^{2}} \end{array}$$
(31)$$\begin{array}{ccc} \sigma_{x,F,m}^{2} & = & \frac{1}{{\left(2\pi \sigma_{w}\right)}^{2}}. \end{array}$$

The expression in (29) is a complex Gaussian function with a TV mean $f_{M,l}^{\left(q\right)}\left(t\right)$ and a complex variance $\sigma_{x,M,l}^{2}$. Note that the complex variance $\sigma_{x,M,l}^{2}$ in (30) is dependent on the slope $k_{M,l}^{\left(q\right)}$ of the Doppler frequency $f_{M,l}^{\left(q\right)}\left(t\right)$ [see (Equation 23)]. The last step is to obtain the spectrogram $S_{H^{\left(q\right)}}\left(f,t\right)$ associated with the *q*th subcarrier by squaring the magnitude of the STFT $X_{H^{\left(q\right)}}\left(f,t\right)$;, i.e.,
(32)$$\begin{array}{ccc} S_{H^{\left(q\right)}}\left(f,t\right) & \approx & |X_{H^{\left(q\right)}}{\left(f,t\right)|}^{2}\;\;=\;\;S_{H^{\left(q\right)}}^{\left(a\right)}\left(f,t\right)+S_{H^{\left(q\right)}}^{\left(c\right)}\left(f,t\right) \end{array}$$
where the functions $S_{H^{\left(q\right)}}^{\left(a\right)}\left(f,t\right)$ and $S_{H^{\left(q\right)}}^{\left(c\right)}\left(f,t\right)$ are called the auto-term and the cross-term of the spectrogram $S_{H^{\left(q\right)}}\left(f,t\right)$, respectively. The auto-term is given by
(33)$$S_{H^{\left(q\right)}}^{\left(a\right)}\left(f,t\right)\approx c_{M}^{2}\;G\left(f,f_{M,l}^{\left(q\right)}\left(t\right),\sigma_{M,l}^{2}\right)\;+\;\sum_{m=1}^{M}c_{F,m}^{2}\;G\left(f,0,\sigma_{F,m}^{2}\right)$$
for $t_{l}<t\leq \;t_{l+1}$, where
(34)$$\begin{array}{ccc} \sigma_{M,l}^{2} & = & \frac{1+{\left(2\pi \sigma_{w}^{2}k_{M,l}^{\left(q\right)}\right)}^{2}}{2{\left(2\pi \sigma_{w}\right)}^{2}} \end{array}$$
(35)$$\begin{array}{ccc} \sigma_{F,m}^{2} & = & \frac{1}{2{\left(2\pi \sigma_{w}\right)}^{2}}. \end{array}$$

The auto-term $S_{H^{\left(q\right)}}^{\left(a\right)}\left(f,t\right)$ in (Equation 33) is an approximation that provides insight into the Doppler power spectrum of the proposed 3D no n-stationary channel model presented in Section 2. This term is real, positive, and consists of a sum of M+1 weighted Gaussian functions. The first Gaussian function, which is due to the moving scatterer $S^{M}$ is weighted by the squared path gain $c_{M}^{2}$ and centered on the approximated TV Doppler frequency $f_{M,l}^{\left(q\right)}\left(t\right)$. The second term of the auto-term $S_{H^{\left(q\right)}}^{\left(a\right)}\left(f,t\right)$ in (Equation 33) is the sum of weighted Gaussian functions, which capture the effect of the M fixed scatterers $S_{m}^{F}$. The weighting factors are the squared path gains $c_{F,m}^{2}$ and each Gaussian function is centered on zero-frequency as the fixed scatterers do not cause Doppler shifts in F2F channels.

The cross-term $S_{H^{\left(q\right)}}^{\left(c\right)}\left(f,t\right)$ of the spectrogram corresponding to the *q*th subcarrier is given by
(36)$$\begin{array}{ccc} S_{H^{\left(q\right)}}^{\left(c\right)}\left(f,t\right) & \approx & \frac{2}{\sigma_{w}\sqrt{\pi }}\;ℜ\{\sum_{n=1}^{M-1}\sum_{\begin{array}{c} m=n+1 \end{array}}^{M}G\left(f,0,\sigma_{x,F,n}^{2}\right)\;G^{*}\left(f,0,\sigma_{x,F,m}^{2}\right)\;H_{F,n}\;H_{F,m}^{*}\\ & & +\sum_{\begin{array}{c} m=1 \end{array}}^{M}G\left(f,f_{n,l}^{\left(q\right)}\left(t\right),\sigma_{x,n,l,M}^{2}\right)\;G^{*}\left(f,0,\sigma_{x,F,m}^{2}\right)\;H_{M}\left(t,\Delta f^{\left(q\right)}\right)\;H_{F,m}^{*}\}. \end{array}$$

The cross-term $S_{H^{\left(q\right)}}^{\left(c\right)}\left(f,t\right)$ in (Equation 36) represents the undesired spectral interference term consisting of M(M+1)/2 components which reduce the resolution of the spectrogram. This term is real but not necessarily positive. The operators ℜ{·} and ${\left(\cdot \right)}^{*}$ denote the real value operator and the complex conjugate operator, respectively. The cross-term in (Equation 36) consists of two terms. The first term of (Equation 36) designates the sum of the components corresponding to the spectral interference caused by the fixed scatterers. The *m*th component of the second term denotes the spectral interference between the moving scatterer $S^{M}$ and the *m*th fixed scatterer $S_{m}^{F}$. The cross-term $S_{H^{\left(q\right)}}^{\left(c\right)}\left(f,t\right)$ in (Equation 36) is dependent on the random phases $\theta_{M}$ and $\theta_{F,m}$ unlike the auto-term $S_{H^{\left(q\right)}}^{\left(a\right)}\left(f,t\right)$. Hence, the cross-term $S_{H^{\left(q\right)}}^{\left(c\right)}\left(f,t\right)$ can be eliminated by taking the average over the random phases, i.e., $E\left\{S_{H^{\left(q\right)}}^{\left(c\right)}\left(f,t\right)\right\}{|}_{\theta_{M},\;\theta_{F,m}}=0$, and thus, $E\left\{S_{H^{\left(q\right)}}\left(f,t\right)\right\}{|}_{\theta_{M},\;\theta_{F,m}}=S_{H^{\left(q\right)}}^{\left(a\right)}\left(f,t\right)$.

The TV mean Doppler shift can be obtained by using the spectrogram as follows
(37)$$B_{H^{\left(q\right)}}^{\left(1\right)}\left(t\right)=\frac{\int_{-\infty }^{\infty }fS_{H^{\left(q\right)}}\left(f,t\right)df}{\int_{-\infty }^{\infty }S_{H^{\left(q\right)}}\left(f,t\right)df}.$$

If the spectrogram $S_{H^{\left(q\right)}}\left(f,t\right)$ in (Equation 37) is replaced by the auto-term $S_{H^{\left(q\right)}}^{\left(a\right)}\left(f,t\right)$, the TV mean Doppler shift $B_{H^{\left(q\right)}}^{\left(1\right)}\left(t\right)$ becomes equal to $B_{f^{\left(q\right)}}^{\left(1\right)}\left(t\right)$, i.e., $B_{H^{\left(q\right)}}^{\left(1\right)}\left(t\right)=B_{f^{\left(q\right)}}^{\left(1\right)}\left(t\right)$.

## 5. Measurements and Numerical Results

In this section, we discuss and compare the TV Doppler power characteristics of our proposed channel model with those of measured CSI data. We will describe the processing of the measured trajectory during the measurements.

### 5.1. Measurement Scenario

To complement the TV Doppler power characteristics of the proposed channel model, measurements have been performed. The CSI data and the trajectory of a pendulum (moving object) have been measured simultaneously. Two laptops have been used for measuring the CSI as Wi-Fi $T_{x}$ and $R_{x}$. An IMU sensor fusion has been used to measure the trajectory of the pendulum. Figure 2 illustrate the measurement scenario in xy and xz planes, respectively. The pendulum was a 3 kg medicine ball, covered with aluminum foil and attached to the ceiling by a rope, and was swinging in a horizontal direction perpendicular to the line-of-sight (LoS). The distance between the ceiling and the center of mass (CoM) of the ball *L* was 1.17 m and the horizontal distance between Wi-Fi $T_{x}$ antenna and the CoM of the ball was 1.5 m. The distance between Wi-Fi $T_{x}$ and $R_{x}$ antennas was 2 m and they had the same height value of 1.18 m. The initial location of the moving scatterer (ball) was the origin. The pendulum displacements $x_{M}\left(t\right)$ and $z_{M}\left(t\right)$ are computed as follows [36]: (38)$$\begin{array}{ccc} x_{{}_{M}}\left(t\right) & = & L\sin\left(\arcsin\left(\frac{x_{\max}}{L}\right)\cos\left(\sqrt{\frac{g}{L}}t\right)\right) \end{array}$$(39)$$\begin{array}{ccc} y_{{}_{M}}\left(t\right) & = & 0 \end{array}$$(40)$$\begin{array}{ccc} z_{{}_{M}}\left(t\right) & = & L\left\{1-\cos\left[\arcsin\left(\frac{x_{{}_{M}}\left(t\right)}{L}\right)\right]\right\} \end{array}$$
where *g* denotes the acceleration of gravity. The parameters $x_{\max}$ and *L* in (38)–(40) were set to 0.55 m and 1.17 m according to Figure 2a,b respectively.

### 5.2. Motion Capturing Using IMU

A MetaMotionR sensor fusion (IMU) [51] was attached to the swinging ball. A smartphone was connected via Bluetooth to control the IMU and log the data files. The IMU was used to record quaternions and linear accelerations during the experiment. Euler angles were computed by using the recorded quaternions to rotate the measured linear accelerations. Next, the raw rotated linear accelerations were smoothed by using quadratic regression provided by the signal analysis toolbox in MATLAB 2019a. After that, the rotated linear accelerations were integrated and double integrated to obtain the velocities and the displacements (trajectories), respectively. Due to measurement errors of the IMU, the velocities and the displacements suffer from linear and quadratic drifts, respectively. To overcome this drift issue, zero-update (ZUPT) algorithms [45] were employed. Since the pendulum motion is periodic, its horizontal and vertical velocities reach zero when the horizontal and vertical accelerations approach their maximum or minimum values. Similarly, the values of horizontal and vertical displacements approach zero values when the velocities tend to their maximum or minimum values. Hence, by searching for the indices corresponding to the local maximum or minimum values of the accelerations, the velocity drift is removed between two consecutive indices. Also, by knowing the indices of the local maximum or minimum values of the drift-eliminated velocities, the drift of the displacement is removed. The source code of the ZUPT algorithm, where the sensors are placed on the toes of a walking person for position tracking, is available online [46]. This algorithm was repeated to also eliminate the drift of the displacement. Figure 3a depicts the TV drift-free horizontal displacements $x_{M}\left(t\right)$ of the captured data from the IMU and the mechanical model of the pendulum in (Equation 38) by using the pendulum parameters shown in Figure 2. The TV drift-free vertical displacements $z_{M}\left(t\right)$ of the captured data from the IMU and the mechanical model of the pendulum in (40) by using the pendulum parameters shown in Figure 2, are depicted in Figure 3b. A minimal error is noticed between the IMU data and the model in the order of centimeters during the whole interval of 15 s.

### 5.3. Capturing CSI Data

The CSI tool in [18] was installed to capture the CSI data (RF signals). Two HP Elitebook 6930p laptops equipped with Intel NIC5300 were used. An Ubuntu 14.04 LTS operating system was installed on both laptops. One laptop was the transmitter station in injector mode and the other laptop was the receiver operating in monitor mode. The carrier frequency $f_{0}$ was set to 5.32 GHz corresponding to channel 64 according to IEEE 802.11n standards [19]. The sampling frequency and the bandwidth were set to 1 KHz and 20 MHz, respectively. TV phase distortions exist due to carrier frequency offset [21,22,23], sampling frequency offset [24,25,26], and packet boundary delay [27,28]. These TV phase distortions were eliminated by using a B2B connection between the transmitter station and the receiver station as described in [35]. Since there was only one RF transmission port in the Wi-Fi $T_{x}$, an RF power splitter ZFSC-2-10G+ from with two output ports was used. One of the output ports was used for the B2B connection and the other one was connected to the transmitting antenna. At the Wi-Fi receiver laptop, one of the ports was used for the B2B connection, and another port was connected to the receiver antenna. The port used for the B2B connection was connected to a 30 dB attenuator. RF cables 141-1MSM+ from Mini-Circuits^®^ were used as well. The processing of the captured CSI data was done by using MATLAB R2019a. Two matrices are stored in a file. One matrix contains the CSI data that corresponds to the captured signal with the fingerprint information associated with the motion of the pendulum and TV phase distortions. The other matrix corresponds to the B2B connection, i.e., it only contains the TV phase distortions. Then, the matrix that contains the fingerprint information and TV phase distortions is divided by the matrix corresponding to the B2B connection in elementwise form. The output matrix resulting from the elementwise division only contains the fingerprint information. After the elementwise division, a highpass filter has been used to reduce the power of zero-frequency components associated with the fixed scatterers and/or the line-of-sight.

Regarding the channel model and its spectrogram, Figure 4a shows the block diagram of the proposed channel model discussed in Section 3 fed with IMU data as inputs and the computation of the spectrogram. Figure 4b shows the block diagram of the proposed channel model discussed in Section 3 fed with the mechanical model as inputs and the computation of the spectrogram. Note that the difference between Figure 4a and Figure 4b is how the trajectories are obtained to feed the channel model. If they are measured using IMU, then the preprocessing mentioned Section 5.2 should be considered before feeding them to the simulator. If they are computed using the expressions in (38)–(40), then they can be fed into the simulator directly. The channel model can be fed with the TV displacements from either the IMU (after applying ZUPT) or the mechanical model presented earlier in Section 5.2, as inputs. The carrier frequency of the simulator $f_{0}$ was set to 5.32 GHz for consistency with CSI measurement scenario. The number of the fixed scatterers M was chosen to be 6. The initial location of the moving scatterer $S^{M}$ and the locations of the Wi-Fi $T_{x}$, Wi-Fi $R_{x}$ were set according to the experiment scenario as presented in Figure 2, i.e., they can be located anywhere, but the distances should be the same as those illustrated in Figure 2. Then, the TV displacements as presented in Figure 3 were added to the initial location of the moving scatterer $S^{M}$. After that, the TV Doppler frequency $f_{{}_{M}}^{\left(q\right)}\left(t\right)$ caused by the moving scatterer $S^{M}$ was computed according to (Equation 21). The path gains of the moving scatterer $S^{M}$ and each fixed scatterer $S_{m}^{F}$ were computed by
(41)$$\begin{array}{cccccc} c_{{}_{M}} & =\sqrt{2\eta } & \; & \mathrm{and} & c_{F,m} & =\sqrt{\frac{2\left(1-\eta \right)}{M}} \end{array}$$
respectively. The parameter η is used to balance the contribution of the fixed and moving scatterers and was set to 0.8. The phases $\theta_{{}_{M}}$ and $\theta_{F,m}$ were generated as realizations of random variables with uniform distribution from −π to π. Next, the STFT $X_{H^{\left(q\right)}}\left(f,t\right)$ for each subcarrier index *q* was computed according to (Equation 28). The window spread parameter $\sigma_{w}$ was set 31.1 ms. Finally, the spectrogram S(f,t) (or $\tilde{S}\left(f,t\right)$ in case of using IMU data as inputs) was computed as the squared magnitude of the sum of the STFT over the subcarriers by the following expression
(42)$$\begin{array}{ccc} S(f,t) & = & {\left|\sum_{q}X_{H^{\left(q\right)}}\left(f,t\right)\right|}^{2}. \end{array}$$

For computing the spectrogram of the recorded CSI as exhibited in Figure 4c, the CTF ${\hat{H}}^{\left(q\right)}\left(t,\Delta f^{\left(q\right)}\right)$ is recorded. Then, the STFT ${\hat{X}}_{H^{\left(q\right)}}\left(f,t\right)$ was computed for each subcarrier *q*. After that the spectrogram $\hat{S}\left(f,t\right)$ is computed according to (Equation 42).

Figure 5a–c exhibit the spectrograms of $\tilde{S}\left(f,t\right)$, S(f,t), and $\hat{S}\left(f,t\right)$ of the channel model with IMU data as inputs, the channel model fed with the mechanical model as inputs, and the recorded CSI data, respectively. It is shown that the TV Doppler power characteristics depicted by the spectrograms $\tilde{S}\left(f,t\right)$, S(f,t), and $\hat{S}\left(f,t\right)$ in Figure 5a–c are fairly similar to each other, respectively. In Figure 5a–c, the Doppler frequency associated with the moving scatterer (pendulum) $S^{M}$ has negative values when the pendulum swings away from the Wi-Fi $T_{x}$ and Wi-Fi $R_{x}$ antennas and has positive values when it swings towards them. The Doppler frequency corresponding to the moving scatterer (pendulum) $S^{M}$ approaches zero values at the time instants in which the moving scatterer reach its local maximum and minimum displacement values [see Figure 3a,b]. Therefore, the speed of the pendulum $v_{{}_{M}}\left(t\right)$ approaches zero values. Thus, the Doppler shift at these instants is zero according to (Equation 22).

Figure 6 depicts the TV mean Doppler shifts ${\tilde{B}}^{\left(1\right)}\left(t\right)$, $B^{\left(1\right)}\left(t\right)$, and ${\hat{B}}^{\left(1\right)}\left(t\right)$ computed from the spectrograms $\tilde{S}\left(f,t\right)$, S(f,t), and $\hat{S}\left(f,t\right)$ using (Equation 37), respectively. There is a good match between ${\tilde{B}}^{\left(1\right)}\left(t\right)$, $B^{\left(1\right)}\left(t\right)$, and ${\hat{B}}^{\left(1\right)}\left(t\right)$. The mean Doppler shifts have negative values at the time instants in which the pendulum (moving scatterer $S^{M}$) swings away from the Wi-Fi $T_{x}$ and Wi-Fi $R_{x}$ antennas. They have positive values when the pendulum swings towards the Wi-Fi $T_{x}$ and Wi-Fi $R_{x}$ antennas. The TV mean Doppler shifts ${\tilde{B}}^{\left(1\right)}\left(t\right)$, $B^{\left(1\right)}\left(t\right)$, and ${\hat{B}}^{\left(1\right)}\left(t\right)$ approach zero values at the moments when the pendulum reaches its local maximum and minimum displacement values. There exists a slight drift in the values of the mean Doppler shift ${\hat{B}}^{\left(1\right)}\left(t\right)$ in between the time instants t≈11.5 s and t≈12.7 s due to the no ise of the measured CSI signal.

For quantitative evaluation, we collected CSI and IMU data for 20 experiments, i.e., K=20. From the collected data measurement, we computed the normalized-mean-square-error (NMSE) $\gamma_{k}$ between the TV mean Doppler shift ${\tilde{B}}_{k}^{\left(1\right)}\left(t\right)$ of the proposed channel model fed with the IMU data as inputs and the TV mean Doppler shift ${\hat{B}}_{k}^{\left(1\right)}\left(t\right)$ of the CSI data according to
(43)$$\gamma_{k}=\frac{\int_{0}^{T_{\mathrm{obs}}}{\left({\tilde{B}}_{k}^{\left(1\right)}\left(t\right)-{\hat{B}}_{k}^{\left(1\right)}\left(t\right)\right)}^{2}dt}{\int_{0}^{T_{\mathrm{obs}}}{\left({\hat{B}}_{k}^{\left(1\right)}\left(t\right)\right)}^{2}dt}$$
for k=1,2,⋯,K, where the parameter $T_{\mathrm{obs}}$ denotes the observation interval which was set to 15 s, i.e., $T_{\mathrm{obs}}=15$ s. Figure 7 depicts the NMSE $\gamma_{k}$ for each experiment. The maximum NMSE belongs to the first experiment and has a value of 0.1829, whereas the minimum NMSE, with a value of 0.0477, belongs to fourteenth experiment. The average NMSE equals 0.0932, and the variance of the NMSE is 0.0013.

## 6. Conclusions

In this paper, we proposed a no n-stationary wideband channel model and its TV Doppler power characteristics when there is a moving object in the 3D space. We derived the TV Doppler shift caused by the moving object in terms of the TV speed, AAOM, EAOM, AAOD, EAOD, AAOA, and EAOA. The TV Doppler characteristics of the proposed channel model were analyzed by using the spectrogram. Furthermore, we provided the approximate solution of the spectrogram of the channel model. We validated the proposed channel model by measuring the trajectory of the moving object using an IMU and calibrated CSI with B2B connection, simultaneously. Then, we fed the channel model with the trajectory data extracted form the IMU. The results showed a good agreement between the measured CSI and the channel model in terms of the spectrogram and the mean Doppler shift. We conclude that the proposed channel model can be used for designing simulation-based HAR systems. For the future, we aim to extend the proposed channel model for human activity recognition by modelling the moving human as multiple moving scatterers.

## Figures and Tables

**Figure 1 sensors-20-01049-f001:**
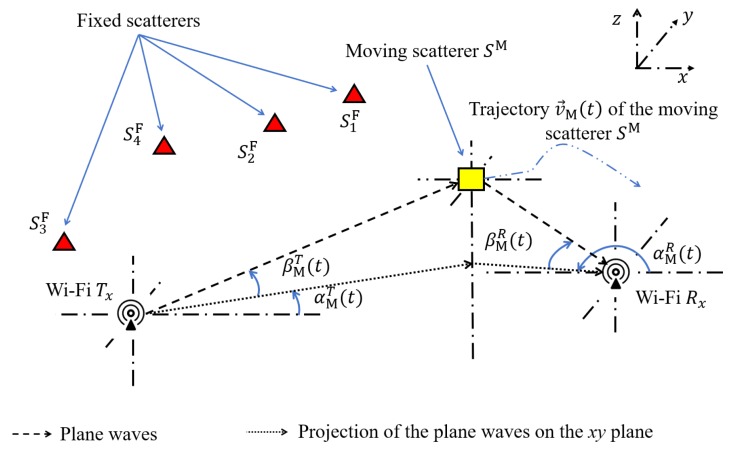
Geometrical model of a 3D multipath propagation scenario with one moving scatterer $S^{M}$ and M fixed scatterers $S_{m}^{F}$, m=1,2,⋯,M.

**Figure 2 sensors-20-01049-f002:**
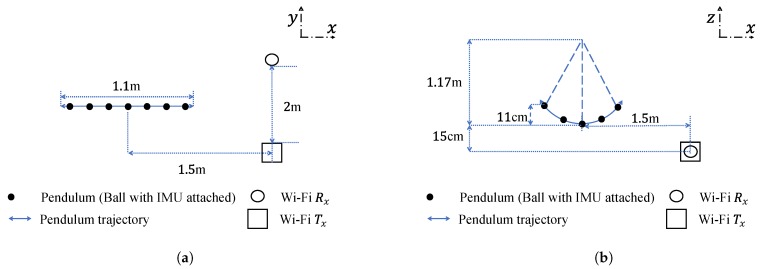
A presentation of the experiment scenario in the (**a**) xy plane and (**b**) xz plane.

**Figure 3 sensors-20-01049-f003:**
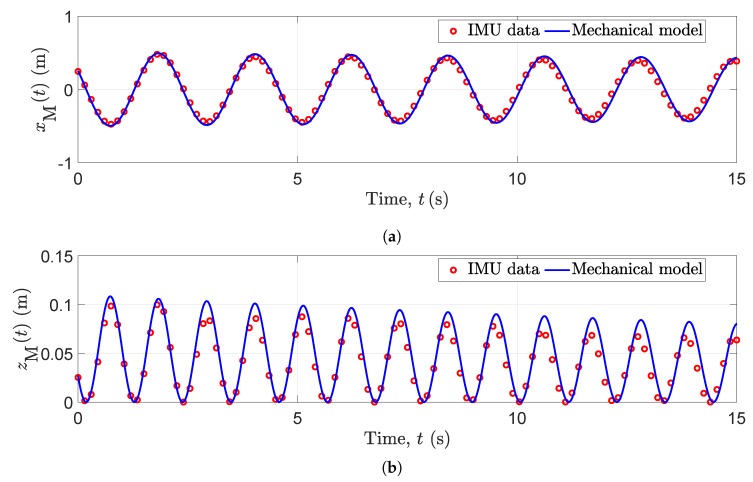
Trajectories of the mechanical model and measured IMU data in (**a**) horizontal direction $x_{M}\left(t\right)$ and (**b**) vertical direction $z_{M}\left(t\right)$.

**Figure 4 sensors-20-01049-f004:**
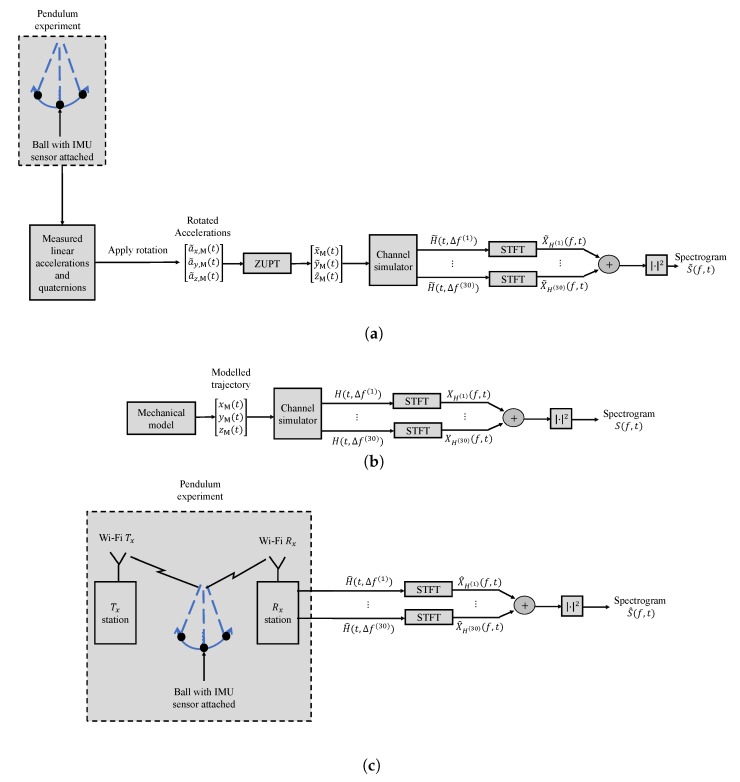
Block diagrams illustrating steps to compute the spectrograms $\tilde{S}\left(f,t\right)$, S(f,t), and $\hat{S}\left(f,t\right)$ of (**a**) the channel model with the IMU data as input, (**b**) the channel model with the trajectories of the mechanical model as inputs, and (**c**) measured CSI data, respectively.

**Figure 5 sensors-20-01049-f005:**
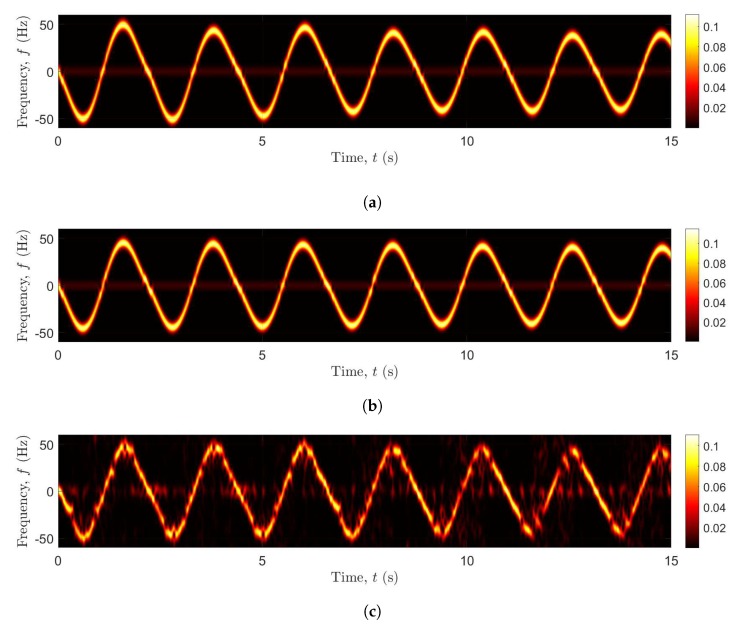
Spectrograms $\tilde{S}\left(f,t\right)$, S(f,t), and $\hat{S}\left(f,t\right)$ (**a**) the channel model with IMU data as inputs, (**b**) the channel model with the mechanical model of the pendulum as inputs, and (**c**) measured CSI, respectively.

**Figure 6 sensors-20-01049-f006:**
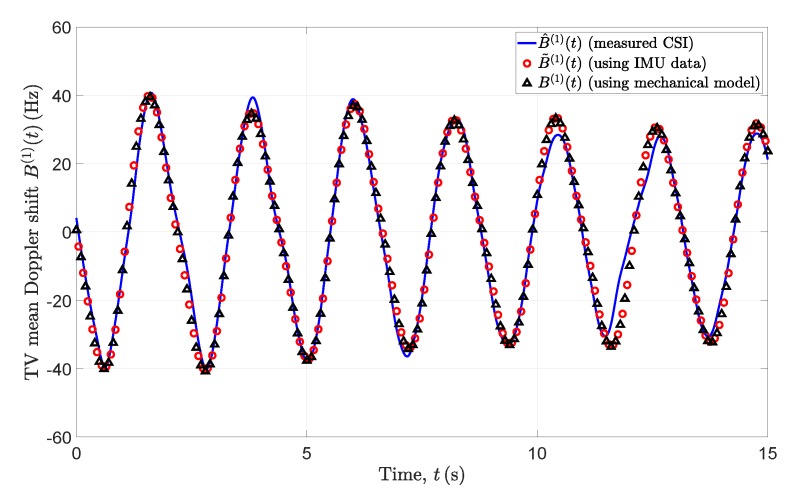
TV mean Doppler shifts ${\tilde{B}}^{\left(1\right)}\left(t\right)$, $B^{\left(1\right)}\left(t\right)$, and ${\hat{B}}^{\left(1\right)}\left(t\right)$ computed from the spectrograms of the channel model with IMU data as inputs, mechanical model as inputs, and the measured CSI, respectively.

**Figure 7 sensors-20-01049-f007:**
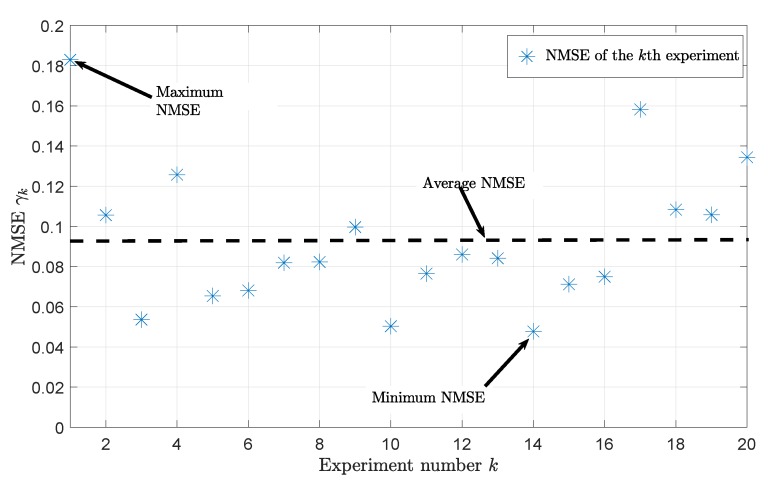
NMSE $\gamma_{k}$ of each experiment computed from the TV mean Doppler shift ${\tilde{B}}_{k}^{\left(1\right)}\left(t\right)$ of the channel model fed with IMU data as inputs and the measured TV mean Doppler shift ${\hat{B}}_{k}^{\left(1\right)}\left(t\right)$ for k=1,2,⋯,K.

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
