# Peer review of "Modelling, Analysis, and Simulation of the Micro-Doppler Effect in Wideband Indoor Channels with Confirmation Through Pendulum Experiments"

_sensors, 2020, doi:10.3390/s20041049_

Round 1

Reviewer 1 Report

In this paper, the authors introduce a 3D nonstationary wideband channel model, which can be used for activity recognition. They also derive expressions for the Doppler shift caused by a mobile scatterer and the spectrogram of the complex channel transfer function. Proposed models are validated using the results of a pendulum experiment. Overall, the paper is very well written. I have only the following minor comments

1."HAR" in pg. 2 is not defined.
2. On pg. 5, line 12: Please remove "of" in "The first term of in (17)".

3.In addition to the agreement shown between the model and the measurements in Figs. 5 and 6, I suggest that the authors increase the number of experiments and give some statistics (e.g., mean-square-error) of the results in a table so that the readers have a better understanding of the accuracy of the proposed models. If possible, comparing the results with that of the existing studies in the same table may also help. Showing the agreement between the model and the measurements only visually, without above mentioned numerical analysis, does not sound technical enough

Author Response

Response to Reviewer 1 Comments

Point 1: ”HAR” in pg. 2 is not defined.

Response 1: Now ”HAR” is defined on page 2, line 33.

Point 2: On pg. 5, line 12: Please remove ”of” in ”The first term of in (17)”.

Response 2: Thank you. We removed ”of” on page 5, line 128.

Point 3: I suggest that the authors increase the number of experiments and give some statistics (e.g., mean-square-error) of the results in a table so that the readers have a better understanding of the accuracy of the proposed models. If possible, comparing the results with that of the existing studies in the same table may also help. Showing the agreement between the model and the measurements only visually, without above mentioned numerical analysis, does not sound technical enough.

Response 3: We agree with the reviewer. More experiments have been conducted and evaluated numerically by using the normalized-mean-square-error between the TV mean Doppler shift of the proposed channel model and the measured TV mean Doppler shift. See the new paragraph added to the revised paper on page 14, including the new Figure 7.

Reviewer 2 Report

The paper proposed a non-stationary wideband channel model and its TV Doppler power characteristics when there is a moving object in the 3D space, which is able to pave the way towards designing simulation-based activity recognition systems. My concerns mainly include:

1) In my opinion, there are some model-based studied in the field of RF-based HAR, such as the Fresnel Zone model. Authors should justify the contribution of this work by comparing the proposed model with existing ones.

2) The proposed model should be evaluation with more experiments, to demonstrate the usefulness of such a model, both qualitatively and quantitatively.

3)Some important references in this field are missing, e.g.,

From Fresnel Diffraction Model to Fine-grained Human Respiration Sensing with Commodity Wi-Fi Devices.

Wi-Fi CSI-based behavior recognition: From signals and actions to activities.

Freesense: a robust approach for indoor human detection using wi-fi signals.

Reviewer 3 Report

Comparing with the most paper I reviewed for MDPI, it is the first high-quality technique paper with clear motivation, theory model, experimental design, and numerical result. 

I would recommend accepting the paper. 

Author Response

Dear Reviewer,

Thank you for your response.

Regards,

Round 2

Reviewer 2 Report

I have no further concerns.